# On the Fast DHT Precoding of OFDM Signals over Frequency-Selective Fading Channels for Wireless Applications

**Kelvin Anoh** [1,*] , **Cagri Tanriover** [2] , **Moisés V. Ribeiro** [3] , **Bamidele Adebisi** [4] **and Chan Hwang See** [5]

1. School of Engineering, University of Bolton, Bolton BL3 5AB, UK
2. Intel Corporation, Hillsboro, OR 97124, USA
3. Department of Electrical Engineering, Federal University of Juiz de Fora (UFJF), Juiz de Fora 36036-900, MG, Brazil
4. School of Engineering, Manchester Metropolitan University, Manchester M1 5GD, UK
5. School Engineering, Edinburgh Napier University, Edinburgh EH10 5DT, UK
* Correspondence: k.anoh@bolton.ac.uk; Tel.: +44-1204-90-3830

**Abstract:** Due to high power consumption and other problems, it is unlikely that orthogonal frequency-division multiplexing (OFDM) would be included in the uplink of the future 6G standard. High power consumption in OFDM systems is motivated by the high peak-to-average power ratio (PAPR) introduced by the inverse Fourier transform (IFFT) processing kernel in the time domain. Linear precoding of the symbols in the frequency domain using discrete Hartley transform (DHT) could be used to minimise the PAPR problem, however, at the cost of increased complexity and power consumption. In this study, we minimise the computation complexity of the DHT precoding on OFDM transceiver schemes and the consequent power consumption. We exploit the involutory properties of the processing kernels to process the DHT and IFFT as a single-processing block, thus reducing the system complexity and power consumption. These also enable a novel power-saving receiver design. We compare the results to three other precoding schemes and the standard OFDM scheme as the baseline; while improving the power consumption efficiency of a Class-A power amplifier from 4.16% to 16.56%, the bit error ratio is also enhanced by up to 5 dB when using a $\frac{1}{2}-$rate error-correction coding and 7 dB with interleaving.

**Keywords:** discrete Hartley transform (DHT); discrete Fourier transform (DFT); orthogonal frequency division multiplexing (OFDM); precoding

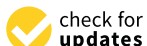



## 1. Introduction

Multi-carrier techniques, such as orthogonal frequency-division multiplexing (OFDM), are used to increase data rates over limited bandwidths in communication systems. This is particularly useful following the increasing demand for higher data rates, COVID-19 pandemic experience and emerging network standards such as 5G and 6G [1]. OFDM uses discrete Fourier transform (DFT) to increase data rates by dividing the limited bandwidths into many, but narrower, numbers of bandwidths. Although the use of DFT in OFDM systems is a mature subject [2,3], the communication technique still finds applications in existing and emerging technologies such as WiFi, WiMAX, DAB, DVB, 4G-LTE and 5G-NR. Due to the smaller bandwidths, OFDM waveforms exhibit a high peak-to-average power ratio (PAPR), and the PAPR has been described as the major drawback in 4G and 5G standards [1]. The PAPR measures the peak-to-average power dissipation of communication systems operating with multi-carrier kernels [2,4,5]. High PAPR drive communication system modules such as the power amplifier (PA) to operate in the saturation region leading to high power consumption [2,6]. In fact, for every 3 dB reduction in PAPR, the PA doubles in its efficiency [7]. Researchers have explored different options, such as clipping, companding and precoding, to minimise the PAPR problem [2,5]. As the PAPR reduction methods have received wide treatments in the literature, this study investigates the computational

inefficiency and power consumption problems introduced by mitigating PAPR problems using precoding schemes, such as discrete-Hartley transform (DHT).

Although several researchers have looked at the PAPR problem in the recent past, we will address the subsequent issues of complexity and power inefficiency associated with reducing the PAPR through DHT precoding of the multi-carrier signals. Due to the high power consumption problems, amongst other things, it is unlikely that OFDM will be included in the forthcoming 6G standard. In fact, the high PAPR problem was the major reason for not including OFDM technology in the uplink of mobile communication standards [8], such as the 4G-LTE. The PAPR problem associated with the DFT-based systems has motivated the use of alternative multi-carrier kernels such as wavelet transform (WT) [9] in IEEE 1901 standard for powerline communication [10] and WiMAX [1] applications. Single-carrier (SC) waveforms, such as DFT-*s*-OFDM, are sometimes used as they can provide 3–5 dB better output power than OFDM waveforms as in the uplink of 5G-NR; the power gain is majorly due to low-PAPR transmission schemes with low modulation orders [3,6]. Due to phase noise problem which increases with the operating frequency [3], it is likely that OFDM will not be included in the future 6G standard which is anticipated to operate in the sub-THz and THz regions [6,11]; note that 5G-NR Release 17 supports carrier frequency up to 71 GHz [12]. As an alternative, SC-waveforms known to provide better PAPR, coverage and power consumption, both in the uplink and the downlink than OFDM [3,6], can be used. In addition, SC-waveform also provides better phase noise than OFDM at higher (e.g., THz) frequencies [6].

Thus, reducing the PAPR of multi-carrier systems is an attractive option for adopting them in the future communications standards such as the 6G standard. Since SC-waveforms provide better PAPR, and the linear precoding of DFT-multi-carrier schemes (e.g., OFDM) could lead to SC-waveforms [13,14], our study spans the computational efficiency in realising linear precoding of OFDM schemes using DHT. As SC-waveforms provide better phase noise, coverage and lower power consumption, minimising the computational complexities of the linearly-precoded OFDM schemes could further reduce the power consumption. Our main contributions in this study are:

- Combining the OFDM and its DHT-precoding kernel into a single-processing block in order to reduce design complexity in the resulting low-complexity precoded-OFDM scheme. The DHT precoding of OFDM converts the multi-carrier waveform into a SC-waveform, thus achieving the above outlined benefits of SC-waveforms;
- Using the involutory property that is shared between DFT and IDFT, and also DHT and IDHT as well as between DFT and DHT to reduce the computational complexity of combining DFT and DHT into a single-processing block, this study achieves a significant computational reduction from $\mathcal{O}(2N \log_2 N)$ to $\mathcal{O}(N)$ only. The proposed model also addresses the power amplification problem that existed in the earlier models;
- We propose a novel receiver-side model of the combined DHT-DFT scheme that preserves the orthogonality of the subcarriers of the OFDM scheme and improves the bit error ratio (BER) performance from irreducible error problems;
- We provide analytical interpretation and quantify the power saving benefits of the improved complexity and the PAPR reduction caused by the DHT-precoding of OFDM and relate these to their impacts on the efficiency of PAs.

We enhance the BER performance of the scheme further by applying error correction coding with interleaving.

## 2. Related Works

In the literature, several precoding schemes exist [15–18]. The transforms used in linear precoding include DFT, DHT, Walsh-Hadarmard transform (WHT), and discrete cosine transform (DCT) [19]. These transforms can be combined into a hybrid to reduce PAPR, such as in the orthogonal multiple-access (NOMA) [20]. The linear precoding schemes are also used in wavelet-OFDM NOMA [21] and wavelet-OFDM PLC [22].

Traditionally, DHT could be used to precode signals in frequency domain, mitigate deep fading/spectral null problem and reduce the PAPR of multi-carrier-based systems, such as in OFDM systems. The transform requires $\mathcal{O}(N^2)$ computational complexity which makes it unattractive for multi-carrier multiplexing schemes. Although there are other PAPR reduction schemes, linear precoding achieves both PAPR reduction as well as BER reduction in OFDM systems such as in [23,24]. The PAPR reduction is due to phase rotation of the OFDM signals while the BER reduction is achieved from spreading of deep fading errors over the entire symbol length. Usually, linear precoding is performed in the frequency domain of OFDM systems before the time domain OFDM processing [25]. Linear precoding also has the potential of reducing the PAPR of OFDM systems to that of an SC waveform when DFT precoding is performed before the time domain OFDM processing [14,26]. The DFT linear precoding of OFDM systems is equivalent to converting the conventional OFDM scheme to SC-FDMA system [13] as in [14]. In terms of BER performance, both DFT and DHT precodings of OFDM, respectively, perform alike. This is because both transforms share similar kernels. It is known that DFT-*s*-OFDM [6], an example of SC scheme, can provide 3–5 dB better output power than OFDM due to low-PAPR transmission schemes with low modulation orders [3]. Other studies have explored the use of DHT as an alternative kernel for DFT in multi-carrier system design with improved PAPR and BER performances [27]. A major limitation of such proposal is that DHT kernel applies to real-signals; however, DFT spans both real and complex signals [23,28].

Although there are a number of studies addressing PAPR reduction using linear precoding of multi-carrier systems, there is increasing interest in reducing the computational complexity associated with using the different precoding schemes, for example [19,20,25,29]. In this study, we focus on the transceiver design of the computationally-reduced DHT precoded OFDM systems. The proposed fast DHT (FHT) precoding scheme for OFDM performs exactly the same as the one reported in [19]. The likely problem would be on the correct method of implementing the receiver-side of the system which we will address in this study.

There are other studies investigating how to reduce the computational burden of DHT precoding OFDM systems [19,23,24] and have not explored the involutory property that we explore in this study. The DHT is computationally expensive on an order of $\mathcal{O}(N^2)$ for an $N$-length OFDM symbol. In other words, as the length of the symbol increases the system power required to drive the DHT computationally expensive kernel increases, significantly. It was demonstrated that DFT and inverse DFT (IDFT) have involutory property in common, as DHT and DFT. In [19,24], it was shown that DHT and IDFT used in precoded OFDM symbols could be combined into a single processing block to enhance both computational and power deficiencies. The combined DHT-IDFT framework gulps over $\mathcal{O}(2N \log_2 N)$ complexity [24] while that of [19] requires $\mathcal{O}(2N)$ complexity which can be reduced to a constant value only.

In this study, we concentrate on the transmitter and receiver-sides design of the system to harness the benefits of the DHT precoding over time-varying fading channel with more than a single-tap delay. The proposed combined FHT-DFT overcomes irreducible errors associated with processing higher number of tap delays. This single-processing block scheme in the transmitter is followed with slightly increased receiver-side complexity due to detection problems, however with a much better BER performance. The PAPR performance reported in [19] is also shown to be preserved since our study spans PAPR reduction including BER improvement and reducing computational complexities. We also show that by incorporating error correction coding with interleaving, the BER performance can be improved further. We derive and interpret the power saving benefits of improved complexity and the PAPR reduction on the efficiency of PAs.

## 3. System Model

We start with the conventional model for designing precoded OFDM systems which involves frequency-domain processing of $N-$input data symbols with linear precoders

before IDFT processing to realize the time-domain contents of an OFDM symbol. Let $\mathbf{P} \in \mathbb{C}^{N \times N}$ be the frequency-domain precoder and $\mathbf{X} \in \mathbb{C}^{N \times 1}$ be the frequency-domain symbol under consideration. Here, $\mathbf{P}$ represents the linear precoder which could be drawn from DFT, DCT, WHT or the DHT precoder under study. Without loss of generality, we assume that the frequency-domain symbol vector $\mathbf{X}$ are made up of elements drawn from a constellation scheme, which can be modulated by a suitable digital modulation scheme (either coherent or non-coherent). The vectorial representation of the frequency-domain precoded OFDM symbol is given by

$$\mathbf{C} = \mathbf{PX} \tag{1}$$

while its time-domain representation is given by

$$\mathbf{c} = \mathbf{F}^{\dagger} \mathbf{C}, \tag{2}$$

where $(\cdot)^{\dagger}$ denotes conjugate transpose, $\mathbf{F}$ is the $N$-sized and normalized DFT matrix usually written as

$$\mathbf{F} = \frac{1}{\sqrt{N}} \begin{bmatrix} 1 & 1 & \cdots & 1 \\ 1 & e^{-j\frac{2\pi}{N}} & \vdots & e^{-j\frac{2\pi(N-1)}{N}} \\ \vdots & \vdots & \ddots & \vdots \\ 1 & e^{-j\frac{2\pi(N-1)}{N}} & \cdots & e^{-j\frac{2\pi(N-1)(N-1)}{N}} \end{bmatrix}, \tag{3}$$

and $j = \sqrt{-1}$. An element of $\mathbf{F}$ can be expressed as $F_{n,k} = \exp\left(-j\frac{2\pi nk}{N}\right) / \sqrt{N}$, $\forall n, k = 0, 1, \cdots, N - 1$. Since $\mathbf{F}$ requires $\mathcal{O}(N^2)$ operations, herein, we have implemented (3) using the computationally efficient fast Fourier transform (FFT) algorithm that requires only $\mathcal{O}(N \log N)$ operations. Here $\mathbf{F}^{\dagger}$ is the IDFT implemented as inverse FFT (IFFT). Both DFT and IDFT are individually drawn from $(\mathbf{F}, \mathbf{F}^{\dagger}) \in \mathbb{C}^{N \times N}$ and form transform conjugate pair. In other words, they satisfy the $\mathbf{F}^{\dagger}\mathbf{F} = \mathbf{F}\mathbf{F}^{\dagger} = \mathbf{I}_N$ property where $\mathbf{I}_N$ is an $N \times N$ identity matrix.

Let $\{h[n]\}_{n=0}^{\ell_h - 1}$ be the channel impulse response (CIR) with $\ell_h$ as the length of the CIR, which covers the worst delay spread, and $h[n] \sim \mathcal{CN}(0, \sigma_h^2)$. Its vectorial representation is given by $\mathbf{h} = [[h[0], h[1], \cdots, h[\ell_h - 1]]^T$, while the corresponding $N$-length frequency domain channel response is $\mathbf{H} = [H_0, H_1, \cdots, H_{N-1}]^T = \mathbf{F}\left[\mathbf{h}^T \, \mathbf{0}_{N-\ell_h}^T\right]^T$, where $\mathbf{0}_{N-\ell_h}$ is an $(N - \ell_h)$-length of a column-wise vector of zeros. Furthermore, we note that $\mathbf{D} = \mathbf{diag}\{\mathbf{H}\} = \mathbf{F}\mathbf{H}_c\mathbf{F}^{\dagger}$, where $\mathbf{H}_c \in \mathbb{C}^{N \times N}$ is the $N$-size circulant convolutional matrix associated with the CIR of the channel, which is denoted by $\mathbf{h}$.

Assume that the coherence time of the channel is longer than the time interval corresponding to one OFDM symbol (i.e., one OFDM block); ideal synchronization at the receiver side; complete channel state information (CSI) is available at the receiver side; the correct use of cyclic-prefix insertion and removal to combat the effect of channel delay spread which could lead to inter-symbol interference (ISI); and the transmission over a multipath fading channel corrupted by additive white Gaussian noise (AWGN). Then, the received symbol associated with the transmission of one OFDM symbol, which is at the input of the DFT at the receiver, is given by

$$\mathbf{y} = \mathbf{H}_c\mathbf{c} + \mathbf{z}, \tag{4}$$

where $\mathbf{z} \in \mathbb{C}^{N \times 1}$ is the circular symmetric AWGN characterized by $\mathbb{E}\{\mathbf{z}\} = \mathbf{0}$ and $\mathbb{E}\{\mathbf{z}\mathbf{z}^{\dagger}\} = \sigma_z^2 \mathbf{I}_N$ (i.e., $\mathcal{CN}(\mathbf{0}, \sigma_z^2 \mathbf{I}_N)$), $\sigma_z^2 = N_0$ as the noise variance, and $\mathbb{E}\{\cdot\}$ represents the expected value operator. It means that in OFDM schemes, the precoded vector, $\mathbf{C}$, is transformed into time-domain using an IDFT matrix $\mathbf{F}^{\dagger}$ and transmitted over the fading channel. Furthermore, the use of the cyclic prefix insertion into the vector $\mathbf{c}$ allows us to

obtain a circulant convolutional matrix. Furthermore, the *N*-sized circulant convolutional matrix, $\mathbf{H}_c$, is obtained after the cyclic prefix removal at the receiver.

The received signal at the output of the DFT is expressed as

$$\mathbf{Y} = \mathbf{Fy} = \mathbf{FH}_c\mathbf{F}^{\dagger}\mathbf{PX} + \mathbf{Z} = \mathbf{DPX} + \mathbf{Z}, \tag{5}$$

where $\mathbf{Z} = \mathbf{Fz}$. Given this system model, we are interested in reducing the complexity introduced by $\mathbf{P}$ especially when DHT is used. Both DFT and DHT share similar kernel properties, except that DHT appeals to real contents. If $\mathbf{P}$ is computationally expensive (such as DHT), such system will not be fit for modern energy-efficient communication systems. Thus, there are progressive steps for minimizing the complexity of combining DHT and DFT as a single processing block [19,24,29–32]. Besides the transmitter side, we are also interested in the receiver design of the precoded system, especially when the DHT precoding model is applied in order to overcome the irreducible error problems that lead to poor BER performance.

The conventional DFT (except when using FFT) requires $\mathcal{O}(N^2)$ operations, similarly DHT requires $\mathcal{O}(N^2)$ computations. Its major limitation is its computational complexity and, of course, hardware resource wastage. Recently, it was shown that combining DHT and DFT as a single processing block could reduce the individually $\mathcal{O}(N^2)$ computations of DHT and DFT, respectively, to $\mathcal{O}(2N)$ operations only [19,29] (see Appendix A for details). In this study, we show that such complexity can be reduced further to $\mathcal{O}(N)$ operations without sacrificing the system performance.

## 4. Proposed Fast DHT Precoding of OFDM Scheme

The proposed precoding combines DHT and DFT blocks into a single but fast processing block as shown in Figure 1.

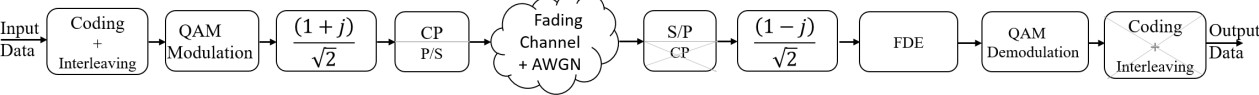

**Figure 1.** System model for DHT precoded OFDM transmission showing the elimination of DHT-IDFT complexities. Note that PS implies parallel-to-serial conversion; similarly, SP represents serial-to-parallel conversion.

### 4.1. The Fast DHT Precoding of OFDM Schemes

Since the FFT implements DFT efficiently reducing the $\mathcal{O}(N^2)$ complexity to only $\mathcal{O}(N\log_2 N)$ assuming the length is a power of two [33], then an efficient computation of DHT is achieved by introducing $\pi/4$-phase shift in each sample of the DFT if DHT can be duly related to DFT. To achieve that, we start with expressing IDFT in terms of DFT; for example, from the involutory relationship $\mathbf{F}^{\dagger}\mathbf{X} = \left(\mathbf{FX}^{\dagger}\right)^{\dagger}$, where $(\cdot)^{\dagger}$ represents complex conjugate operator. The DHT also benefits from the property discussed in [34], in which the authors showed that DHT and DFT are linearly related as $\mathbf{W} = \mathbf{F}(1+j)$, where $\mathbf{W}$ is the *N*-sized and normalized DHT matrix. In this work, we adopt $\mathbf{W} = \mathbf{F}(1+j)/\sqrt{2}$, where $\mathbf{W}$ is the *N*-sized and normalized DHT matrix. The normalization is considered to avoid power amplification that will be induced at the receiver assuming that the DHT was to remain $\mathbf{W} = \mathbf{F}(1+j)$ due to $\mathbf{W}^{\dagger}\mathbf{W} = \mathbf{F}^{\dagger}\mathbf{F}(1+j)(1-j) = 2\mathbf{I}_N$. However, with the normalisation factor $\sqrt{2}$ introduced such that $\mathbf{W} = \mathbf{F}(1+j)/\sqrt{2}$, then $\mathbf{W}^{\dagger}\mathbf{W} = \mathbf{F}^{\dagger}\mathbf{F}(1+j)(1-j)/2 = \mathbf{I}_N$. Consequently, the frequency-domain precoding of the OFDM symbol in (1) can be expressed as

$$\mathbf{C} = \mathbf{F}(1+j)/\sqrt{2}\mathbf{X} = 1/\sqrt{2}(\mathbf{FX} + j\mathbf{FX}). \tag{6}$$

Considering (6) as the DHT precoded signal in the frequency domain, the time-domain vectorial representation of the OFDM symbol after passing through IDFT block becomes

$$\mathbf{c} = \mathbf{F}^{\dagger}\mathbf{C} = \mathbf{F}^{\dagger}(1/\sqrt{2})(\mathbf{FX} + j\mathbf{FX}) = \frac{1+j}{\sqrt{2}}\mathbf{X}. \tag{7}$$

Note that (7) is generated at the transmitter. Clearly, (7) shrinks the complexity of DHT-IDFT block better than (A7) and, as a consequence, only a constant multiplication of $(1+j)/\sqrt{(2)}$ is required. These are comparatively summarised in Table 1.

**Table 1.** Complexity comparison of LDHT [24], OFDM and proposed modified receiver-side FHT (labelled FHT).

| Model | Transform (Tx) | | Transform (Rx) | |
|---|---|---|---|---|
| | **Multiply** | **Adds** | **Multiply** | **Adds** |
| OFDM | $N\log_2 N$ | $N\log_2 N$ | $N\log_2 N$ | $N\log_2 N$ |
| LDHT | $2N-2$ | $N-2$ | $2N\log_2 N$ | $2N\log_2 N$ |
| FHT | 1 | 0 | 1 | 0 |

We showed in Section 3 that at the receiver, the conventional low-complexity DHT (LDHT) LDHT reduces the $\mathcal{O}(N^2)$ complexity of DHT and $\mathcal{O}(N\log_2 N)$ of DFT to $\mathcal{O}(2N)$; however, the computational cost for low-power communication devices such as the mobile phones is still high; this must be reduced. We state that, instead of detecting with $\mathbf{G}^{\dagger}$ (see Appendix A) which requires $\mathcal{O}(N^2)$ complexity at the receiver, only a $(1-j)/\sqrt{2}$ constant operation of the form

$$\hat{\mathbf{X}} = \frac{1-j}{\sqrt{2}}\mathbf{c} = \frac{(1-j)(1+j)}{2}\mathbf{X} = \mathbf{X} \tag{8}$$

is required. Notice that (8) assumes $\mathbf{H}_c = \mathbf{I}_N$ and $\mathbf{z} = 0$ and shows that disregarding equalization, our proposal simplifies both the transmitter and the receiver complexities to only $\frac{\pi}{4}$-phase rotation and amplification by $\sqrt{(|1|^2 + |j|^2)} = \sqrt{2}$ of the vector $\mathbf{X}$. Furthermore, the amplification factor is easily handled to avoid power amplification. This reduces the complexity of performing $\mathcal{O}(N^2)$ of the DHT and $\mathcal{O}(N\log_2 N)$ of the DFT at both the transmitter and the receiver to a $(1 \pm j)/\sqrt{2}$ constant operation.

Moreover, our proposal preserves the BER performance to that of SC-waveform (or DFT-precoded OFDM) as no amplitude distortion is induced while PAPR is significantly reduced because $\mathbf{c}$ is a rotated version of $\mathbf{X}$. Now, from (7), the PAPR at the output of the complexity-shrinked DHT-IDFT block, i.e., $\mathbf{c}$ which is the time-domain content of the signal becomes

$$\text{PAPR} = \frac{P_{\max}}{P_{\text{mean}}} = \frac{\max_i |c_i|^2}{\frac{1}{N}||\mathbf{c}||^2}, \ i = 0, 1, \cdots, N-1, \tag{9}$$

where $|\cdot|$ and $||\cdot||$ represent absolute value and norm operators, respectively. The cumulative complementary density function (CCDF) of the PAPR is found as $Prob\{\text{PAPR} > \gamma\} = 1 - \left[1 - e^{(-\gamma)}\right]^{\alpha N}$, where $\gamma$ is the peak amplitude of $c_i, i = 0, 1, \cdots, N-1$, $\alpha$ is a fitting correction parameter given as $\alpha = 1$ when $N$ is small and $\alpha = 2.8$ when $N$ is large [35].

### 4.2. Power Saving from Reducing Complexity and PAPR

Considering Class-A PAs which are the most linear amplifiers with maximum efficiency of 50% [36], its power efficiency in terms of PAPR can be expressed as

$$\eta = \frac{0.5}{\text{PAPR}},\tag{10}$$

where $\eta = P_{\text{out,ave}}/P_{\text{DC}}$, $P_{\text{out,ave}}$ is the average output power and $P_{\text{DC}}$ is the amount of power consumed by the PA regardless of the input power. From (10), the efficiency of a PA operating with an OFDM scheme having 512-subcarriers whose PAPR is is 12.3 dB ($\approx 16.98$) at $10^{-4}$ probability ([36] (Table I), Table 1) can be found as $\eta = 0.5/16.98 = 2.94\%$. Such low efficiency of PAs necessitates PAPR reduction. We saw in (10) that every 3 dB reduction in the PAPR of OFDM symbols doubles the amplifier efficiency [7]. Thus, applying DHT precoding to OFDM scheme which reduces the PAPR from $\text{PAPR}_{\text{old}}$ to $\text{PAPR}_{\text{new}}$ will cause $\text{PAPR}_{\text{saving}} = \text{PAPR}_{\text{old}} - \text{PAPR}_{\text{new}}$. Combining $\eta = P_{\text{out,ave}}/P_{\text{DC}}$ and (10), $P_{\text{DC}} = 2P_{\text{out,ave}} \times \text{PAPR}$ so that the power saving becomes $P_{\text{saving}} = 2P_{\text{out,ave}} \times \text{PAPR}_{\text{saving}}$.

While the power saving achieved from reducing the FFT/IFFT complexity could be small (usually in µW) compared to the one realised from PAPR reduction [7], the overarching reductions are significant for portable devices and even more for the Internet of Things (IoT) devices of today. Considering a fixed-point digital-signal-processor (DSP), the energy consumption per $N$-point FFT/IFFT [36] can be expressed as

$$\begin{aligned}\text{Energy} &= 415.8\frac{\text{pWs}}{\text{cycle}}\left[306 + 5\frac{N}{2}\log_2\left(\frac{N}{2}\right)\right]\text{cycle} \\ &= 415.8\left[306 + 2.5\left\{2\frac{N}{2}\log_2\left(\frac{N}{2}\right)\right\}\right]\text{pWs}\end{aligned}\tag{11}$$

where $415.8\frac{\text{pWs}}{\text{cycle}}$ is the energy consumed by the fixed-point DSP per cycle. We know that $2\mathcal{O}(N/2) = \mathcal{O}(N)$ operations, in the same way, $2\mathcal{O}(N/2\log_2(N/2)) = \mathcal{O}\{N\log_2(N)\}$ operations. It follows that (11) can be rewritten as

$$\begin{aligned}\text{Energy} &= 415.8[306 + 2.5\{N\log_2(N)\}]\text{pWs} \\ &= [127.23 + 1.04N\log_2(N)]\ (\text{nJ}).\end{aligned}\tag{12}$$

Since the $\mathcal{O}(N\log_2 N)$ computational complexity is responsible for the energy consumption in (12), then from (7), the $\mathcal{O}(N)$ computational complexity would lead to

$$\text{Energy}/N\text{-Point} = \left[\frac{127.23}{\log_2(N)} + 1.04N\right]\ (\text{nJ}).\tag{13}$$

Notably, comparing (12) and (13) it can be found that the energy consumed by the conventional DHT-precoding of OFDM scheme is significantly reduced—when considering the energy costs of DHT and DFT as individual processing blocks.

### 4.3. Receiver Design Considering Fading Channel

In this section, we discuss the receiver side in order to realise the full system-wise design. Recall the received signal model in (5) with the characteristic channel response. To recover the transmitted symbols, suitable equalization scheme must be applied at the receiver to minimise the error-floor in the recovered signals. Such treatment has been missing in earlier studies. To compensate the effect of the channel, let $\mathbf{E}_c$ be the zero-forcing equaliser so that recovered signal be expressed as

$$\hat{\mathbf{X}} = \mathbf{E}_c\mathbf{Y} = \mathbf{E}_c\mathbf{DPX} + \bar{\mathbf{Z}}\tag{14}$$

where $\bar{\mathbf{Z}} = \mathbf{E}_c\mathbf{Z}$. The equaliser should be able to remove the effect of the channel response when applied to the received signal $\mathbf{Y}$. If we let $\mathbf{E}_c = \mathbf{D}^{-1}$ (i.e., equalisation based on the

zero-forcing criterion in the frequency-domain), then the impact of the channel on the signal could be removed; however, the noise is scaled by the $\mathbf{E}_c = \mathbf{D}^{-1}$ term. An OFDM scheme precoded with DHT would face the same equalisation problem. As a result, the receiver must be designed in such a way to overcome this problem. To do that, we formalise the receiver-side as follows:

Let the originally transformed signals be defined as $\mathbf{c} = \mathbf{Q}_T\mathbf{X} \in \mathbb{C}^{N\times1}$, where $\mathbf{Q}_T$ is the transmitter-based transform, such as the IDFT in the conventional OFDM scheme. With respect to (5), the received signal can be expressed as

$$\mathbf{y} = \tilde{\mathbf{y}} + \mathbf{z} = \mathbf{H}_c\mathbf{Q}_T\mathbf{X} + \mathbf{z}. \tag{15}$$

where $\tilde{\mathbf{y}} = \mathbf{H}_c\mathbf{Q}_T\mathbf{X}$. At the receiver let $\mathbf{Q}_R \in \mathbb{C}^{N\times1}$ denote the receiver-based transform, it follows that

$$\mathbf{Y}' = \mathbf{Q}_R\mathbf{y} = \mathbf{Q}_R\mathbf{H}_c\mathbf{Q}_T\mathbf{X} + \mathbf{Q}_R\mathbf{z} \tag{16}$$

where $\mathbf{Z} = \mathbf{Q}_R\mathbf{z}$ is the AWGN scaled by $\mathbf{Q}_R$. Usually in communication systems design, the receiver and transmitter transforms form a unitary pair (i.e., $\mathbf{Q}_T\mathbf{Q}_R^\dagger = \mathbf{Q}_R\mathbf{Q}_T^\dagger = \mathbf{I}$). In OFDM scheme, $\mathbf{Q}_T$ and $\mathbf{Q}_R$ are DFT pair, thus, the received signal in (16) qualifies to be recovered using zero-forcing equalization (ZFE) since $\mathbf{Q}_R\mathbf{H}_c\mathbf{Q}_T$ is a diagonal matrix [24]. In that case, the received symbol can be expressed as

$$\hat{\mathbf{X}} = (\mathbf{Q}_R\mathbf{H}_c\mathbf{Q}_T)^{-1}\mathbf{Y}' = \mathbf{X} + \bar{\mathbf{Z}} \tag{17}$$

where $\bar{\mathbf{Z}} = (\mathbf{Q}_R\mathbf{H}_c\mathbf{Q}_T)^{-1}\mathbf{Q}_R\mathbf{z}$. Because $\mathbf{Q}_T$ and $\mathbf{Q}_R$ are DFT pair and enable $\mathbf{X}$ to be recovered in a single-tap equalization (STE) since $(\mathbf{Q}_R\mathbf{H}_c\mathbf{Q}_T)$ is a diagonal matrix, our goal is to explore the property of the FHT-OFDM scheme to ensure STE at the receiver. From the foregoing discussion, we have noted that $\mathbf{D} = \mathbf{Q}_R\mathbf{H}_c\mathbf{Q}_T$ is a diagonal matrix because $\mathbf{Q}_R = \mathbf{F}$ and $\mathbf{Q}_T = \mathbf{F}^\dagger$ and, as a consequence, $(\mathbf{Q}_R\mathbf{H}_c\mathbf{Q}_T)^{-1} = \mathbf{D}^{-1}$.

Following the same line of argument for the FHT-precoded OFDM scheme under consideration and aiming to reduce the computational complexity of the receiver, we suggest $\mathbf{Q}_R = \mathbf{F}$ and $\mathbf{Q}_T = \mathbf{F}^\dagger\mathbf{W}$, respectively, be the transmitter-based and receiver-based FHT-precoded OFDM systems, respectively. In other words, we take advantage of the existing relationship between $\mathbf{F}$ and $\mathbf{W}$. In this regard, the received signal in the frequency-domain is expressed as

$$\begin{aligned}\mathbf{Y} = \tilde{\mathbf{Y}} + \mathbf{Z} &= \mathbf{Fy} = \mathbf{F}\tilde{\mathbf{y}} + \mathbf{Fz} = \mathbf{F}\mathbf{H}_c\mathbf{F}^\dagger\mathbf{WX} + \mathbf{Fz} \\ &= \mathbf{DWX} + \mathbf{Z}.\end{aligned} \tag{18}$$

Recall that from (6), both DHT and DFT are related by involutory property [34], which involves only a $\pi/4$-phase rotation [32].

Applying the frequency-domain equalisation based on the zero forcing (ZF) criterion, we can thus express an estimate of the transmitted symbol as

$$\begin{aligned}\hat{\mathbf{X}}_{ZF} &= (\mathbf{Q}_R\mathbf{H}_c\mathbf{Q}_T)^{-1}\mathbf{Y} \\ &= \mathbf{E}_{ZF}\mathbf{Y} \\ &= (\mathbf{DW})^{-1}(\mathbf{DWX} + \mathbf{Z}) \\ &= \mathbf{X} + \mathbf{W}^\dagger\mathbf{D}^{-1}\mathbf{Z}.\end{aligned} \tag{19}$$

where $\mathbf{E}_{ZF} = (\mathbf{WD})^{-1} = \mathbf{W}^{\dagger}\mathbf{D}^{-1}$ is the ZF-based criterion FDE. The SNR for $k$-th element of $\hat{\mathbf{X}}_{ZF}$ is defined by

$$\gamma_k \triangleq \frac{\sigma_{X_k}^2}{\sigma_z^2 \sum_{i=0}^{N-1} \frac{1}{|H_i|^2}} \tag{20}$$

where $k = 0, \ldots, N-1$, and $X_k$ and $H_k$ are the $k$-th elements of $\mathbf{X}$ and $\mathbf{H}$, respectively. Note that (20) states the power of the additive noise is a harmonic mean, regardless of the subchannel index. In other words, the maximum achievable data rates is attained when the total transmission power is equally shared among the subcarriers, which is a very interesting result.

Applying the frequency-domain equalisation based on the minimum mean square error (MMSE) criterion, we can thus express an estimate of the transmitted symbol as

$$\begin{aligned}
\hat{\mathbf{X}}_{MMSE} &= \mathbf{W}^{\dagger}\mathbf{E}_{MMSE}\mathbf{Y} \\
&= \mathbf{W}^{\dagger}\mathbf{E}_{MMSE}(\mathbf{DWX} + \mathbf{Z}) \\
&= \mathbf{W}^{\dagger}\mathbf{M}_1\mathbf{DWX} + \mathbf{W}^{\dagger}\mathbf{M}_1\mathbf{Z}
\end{aligned} \tag{21a}$$

where

$$\mathbf{M}_1 = \mathbf{R}_{X_{DHT}}\mathbf{D}^{\dagger}\left(\mathbf{D}\mathbf{R}_{X_{DHT}}\mathbf{D}^{\dagger} + \sigma_z^2\mathbf{I}\right)^{-1} \tag{21b}$$

$$\mathbf{R}_{X_{DHT}} = \mathbb{E}\{(\mathbf{x}_{DHT})^{\dagger}\mathbf{x}_{DHT}\} = \mathbb{E}\{(\mathbf{DWX})^{\dagger}\mathbf{DWX}\}. \tag{21c}$$

If we rewrite (21a) as $\hat{\mathbf{X}}_{MMSE} = \mathbf{AX} + \mathbf{BZ}$, then the SNR in the $k$-th subchannel is defined as follows:

$$\gamma_k \triangleq \frac{\sum_{i=0}^{N-1} |a_{k,i}|^2 \sigma_{X_i}^2}{\sum_{i=0}^{N-1} |b_{k,i}|^2 \sigma_{z_i}^2} \tag{22}$$

where $a_{k,i}$ and $b_{k,i}$ are the values of $A(k,i)$ and $B(k,i)$ elements of matrices $\mathbf{A}$ and $\mathbf{B}$, respectively.

It is important to emphasise that a comparison between both frequency domain equalisers shows the FDE based on the ZF criterion is strongly recommended because the MMSE-based equaliser can demand a prohibitive computational complexity.

In Figure 1, note that since $\mathbf{F}^{\dagger}$ and $\mathbf{W}$ are combined into a single processing block at the transmitter, thus only one $\frac{1+j}{\sqrt{2}}$ constant operation is required. Similarly, since $\mathbf{F}$ and $\mathbf{W}^{\dagger}$ are also combined into a single processing block at the receiver, only the $\frac{1-j}{\sqrt{2}}$ computation is required. From (18) and (19), we state that the receiver requires to performs one DFT as shown in Figure 2 before the ZFE due to $\mathbf{W}^{\dagger} = \mathbf{F}^{\dagger}(1-j)/\sqrt{2}$ term in (19). It follows therefore that, by the DFT operation before FDE block in Figure 2, $(\mathbf{FD})^{-1}$ is a matrix constituted by constant elements during the coherence time of the channel. In this case, the complexity of the receiver is negligible since only $\frac{1-j}{\sqrt{2}}$ computation is required. However, with Figure 2 which requires two DFT operations, then the complexity becomes $2\mathcal{O}(N\log_2 N)$ as shown in Table 1.

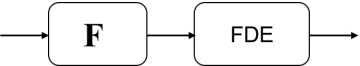

**Figure 2.** Receiver side showing method of implementing the frequency-domain equalisation detection of FHT-OFDM scheme.

## 5. Results and Discussion

In this section, we present the performance of the proposed low-complexity DHT-precoding scheme in comparison to other precoding schemes. To start with, we generated $N = 512$ input signals in MATLAB. Considering a 1/2-rate convolutional coded symbol, that is modulated using 16-QAM, we realize the frequency-domain symbols as shown in Figure 1. The symbol is passed over the proposed FHT-OFDM block $(1 + j)/\sqrt{2}$ which performs DHT-OFDM precoding as a single processing block. The linear precoding converts the conventional multi-carrier waveform to SC-waveform suitable for overcoming the power consumption, phase noise at sub-THz frequencies and coverage problems of standard OFDM scheme. We have not considered the performance of the front-end (that includes the 4G-LTE, 5G-NR, sub-THz or THz frequencies) of the system model as this is outside the scope of our study; for recent studies on this, see [3,6]. With CP included, the output of the FHT-OFDM scheme is transmitted over time-domain Rayleigh fading channel corrupted by AWGN. It is assumed that a perfect channel state (CSI) exists at the receiver and that the receiver is perfectly synchronized to the transmitter. At the receiver, the CP is removed, the signal passed through the proposed low complexity IDHT-FFT receiver $(1 - j)/\sqrt{2}$. As shown in Figure 2, the received signal is passed through DFT as discussed in Section 4.3 before FDE. Afterwards, the resulting signal is demodulated using 16-QAM demodulator. DHT-based OFDM [23] allows single-tap equalizer and the LDHT in [24], an example of DHT-based OFDM, achieves full diversity at 7% complexity increase from equalization which is completely overcome in our model since it requires only a $(1 + j)/\sqrt{2}$ constant operation. For fair comparisons, we chose the earlier LDHT in [19,29] with standard OFDM scheme as the baseline.

### 5.1. Complexity Reduction

The complexity advantages of our model are compared with that of LDHT discussed in [24] and OFDM scheme in Table 1 for transmitter and receiver sides; our model achieves the least complexity among all schemes. For example, in Figure 3, it can be seen that while OFDM scheme has the highest complexity for the increasing number of subcarriers starting from $N = 32$ to $N = 4096$, the proposed complexity-reduced model achieves the lowest complexity for all the three cases at the transmitter. In addition, considering the receiver-side as shown in Figure 4, OFDM achieves the lowest complexity among the three schemes while both the proposed and the earlier LDHT discussed in [24] achieve comparable complexity.

### 5.2. PAPR Performance of Proposed Model

In the foregoing discussions, we have enumerated in Section 1 that PAPR problem is the inhibiting factor for not including OFDM in the uplink of mobile communications standards such 4G-LTE and 5G-NR. Alternatively, precoded OFDM realising SC-waveforms using DFT-*s*-OFDM schemes are used due to the advantages of increased coverage and power consumption. Such scheme bestows $N \log_2 N$ complexity due to the DFT-precoding and another $N \log_2 N$ due to OFDM plus the CP length of up to $0.25N$. With the future 6G standard where the operation frequency will be in the THz region, phase noise problem will also be pronounced. Linear precoding of OFDM with DHT converts the multi-carrier waveform akin to SC-waveform, hence overcoming all the problems inherent with the conventional OFDM scheme including the computational complexities of the associated DFT-*s*-OFDM schemes. In terms of PAPR problem which exacerbates the power consumption of the system, it can be seen in Figure 5, that DHT linear precoding reduces the PAPR of standard OFDM system by 6.0 dB. This result agrees with the earlier proposal discussed in [19,29]. Using (9), the PAPR result of the proposed LDHT in Figure 5 perfectly agrees with earlier LDHT [19] and outperform DCT, FWHT precoding and standard OFDM schemes by 3.8 dB, 5.8 dB and 6.0 dB, respectively; it performs similar to DFT precoded OFDM. These results make it quite attractive for DHT precoding to be included in the

future communication system standards, such as 6G, due to low power consumption and reduced computational efficiency for battery-laden systems including IoT devices.

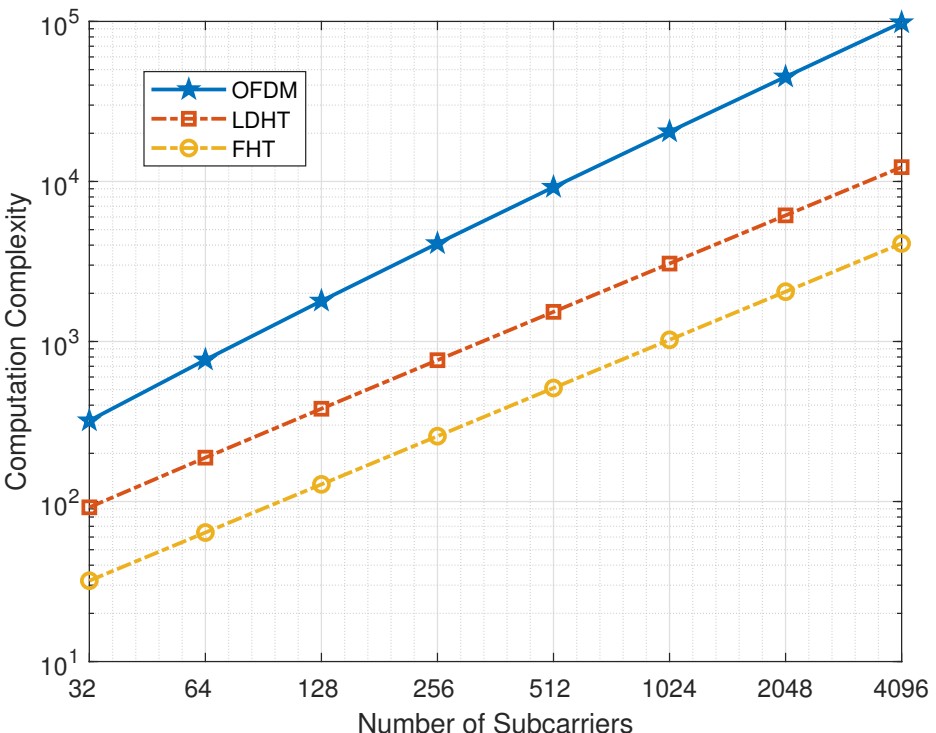

**Figure 3.** Transmitter-side performance comparisons of complexities of the proposed fast DHT-IDFT and other schemes.

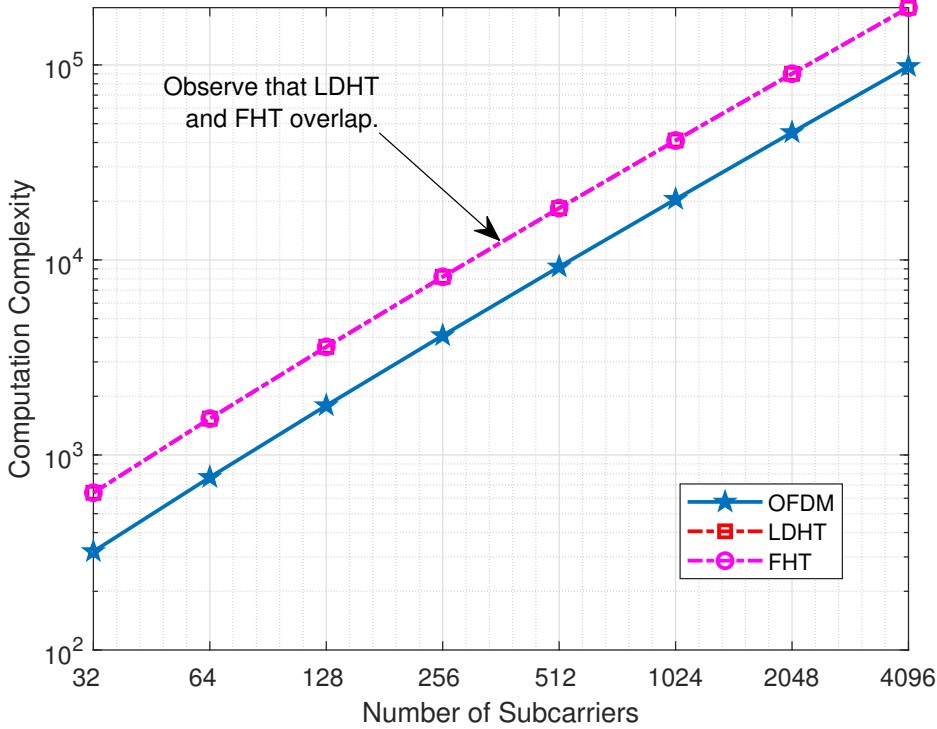

**Figure 4.** Receiver-side performance comparisons of complexities of the proposed fast DHT-IDFT block and other schemes.

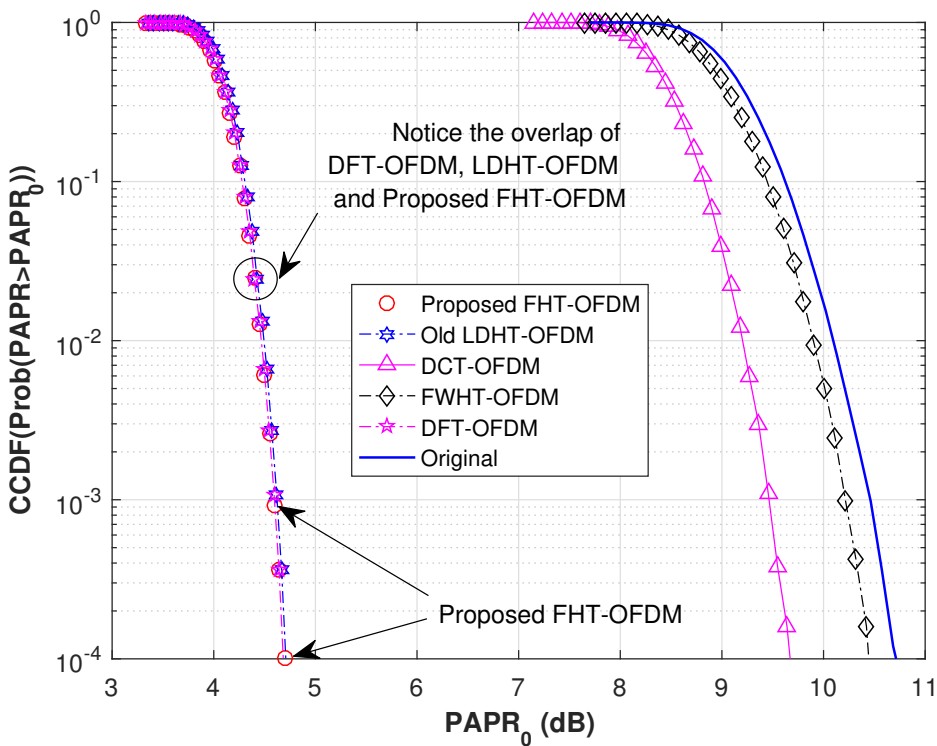

**Figure 5.** Comparison of the PAPR of the proposed LDHT scheme with other OFDM precoding schemes for $N = 512$ subcarriers.

### 5.3. Power Saving Benefits

The power saving benefits of the proposed precoding scheme are two fold; one relating to PAPR reduction and the other relating to complexity reduction. In terms of complexity reduction, we compare the energy consumption derived in (12) and (13) as shown in Figure 6 with that of standard OFDM scheme as the baseline. It can be observed that the energy consumption of the proposed low computation complexity improves the earlier low-complexity DHT precoding model by 74% at $N = 32$ and 67% at $N = 4096$. From Section 4.2 and the PAPR saving in Figure 5, the power saving induced by the DHT precoding scheme can be found as $P_{\text{saving}} = 2P_{\text{out,ave}} \times \text{PAPR}_{\text{saving}} = 2 \times 6 \times 6 = 72$ dB, where $P_{\text{out,ave}}(\text{dB}) = 10 \log_{10}(4W/1W) = 6$dB. Note that $P_{\text{out,ave}} = 4$ watts for portable devices [36]. From (10), the PA efficiency due to PAPR reduction increases from $\eta = 0.5/12.0226 \approx 4.16\%$ to $\eta = 0.5/3.02 \approx 16.56\%$.

### 5.4. BER Performance with FEC Only

In Figure 1, we use 16-QAM to modulate $N = 512$ half-rate convolutionally coded OFDM signals consistent with INMARSART standard as discussed in [37,38]. Afterwards, we combine the DHT-IFFT block to precode OFDM symbols and later add CP. Note that the proposed low complexity DHT scheme performs precoding and IFFT together in a single block using $(1 + j)/\sqrt{2}$ with the added advantage of reduced complexity. Although it was conjectured that the achieved PAPR reduction compared to standard OFDM scheme was achieved due to the transform being sparse [23,24], we observed that the achieved PAPR reduction was due to the conversion of OFDM to a kind of SC-FDMA scheme by DHT precoding. This is a form of SC-waveform with low computational complexity including the potential for the tripartite advantages of reduced power consumption, coverage and phase noise improvements. This provides great advantage in using the proposed LDHT OFDM scheme in terms of significant PAPR reduction, BER performance without loss including added bonus of reduced transmitter and receiver complexities as shown in Table 1. Additionally, with flat fading as shown Figure 7, we observe in Figure 8 that all the

precodings perform similarly as the conventional OFDM scheme operated without precoding. The proposed LDHT achieves similar BER and PAPR performances as the earlier LDHT [19,23,29], with the added advantage of reduced number of computations. These BER performance results are only true when there is a mild-fading channel environment (e.g., $\ell_h = 1$) as shown in Figure 7. However, in a harsh multipath fading environment (e.g., $\ell_h = 3$) as shown in Figure 9, the precoded OFDM schemes outperform the conventional OFDM and old-DHT schemes by 7 dB as shown in Figure 10. We note that the old-DHT scheme suffers from loss of orthogonality leading to poorer BER performance (overlapping with conventional OFDM system) amongst all the linearly precoded systems in Figure 10. Furthermore, notice that 1/2-rate convolutional coding further improves the system performance such that the precoded-OFDM having convolutional coding achieves about 5 dB performance gain compared to the precoded-OFDM without forward error correction coding (FEC). In addition, the 1/2-rate FEC enables the precoded OFDM systems to achieve 12 dB gain better than conventional OFDM system and old-DHT precoding scheme.

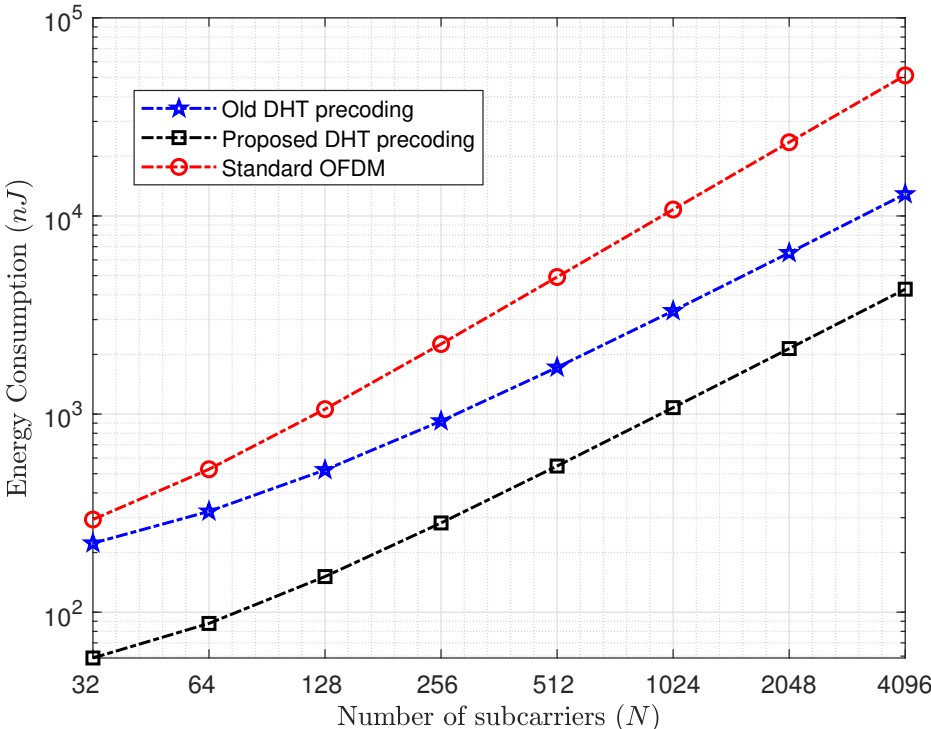

**Figure 6.** Comparison of energy consumption per *N*-point FFT/IFFT of OFDM, DHT-precoded OFDM and OFDM.

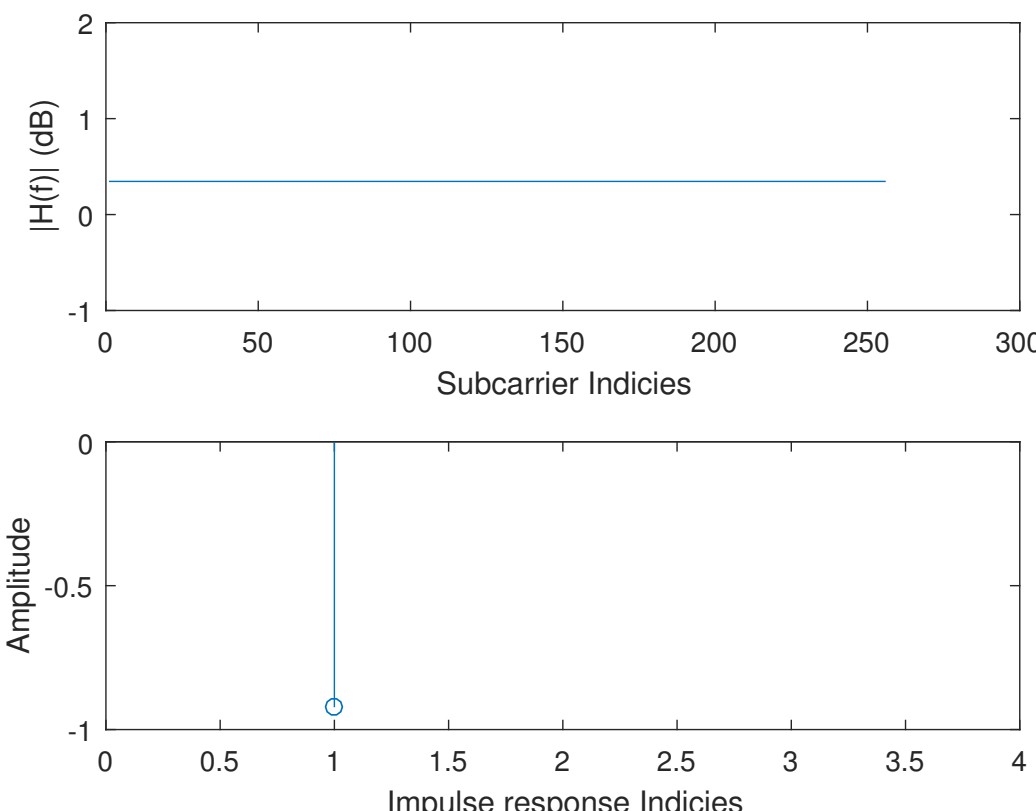

**Figure 7.** Channel and impulse responses ($\ell_h = 1$ channel tap).

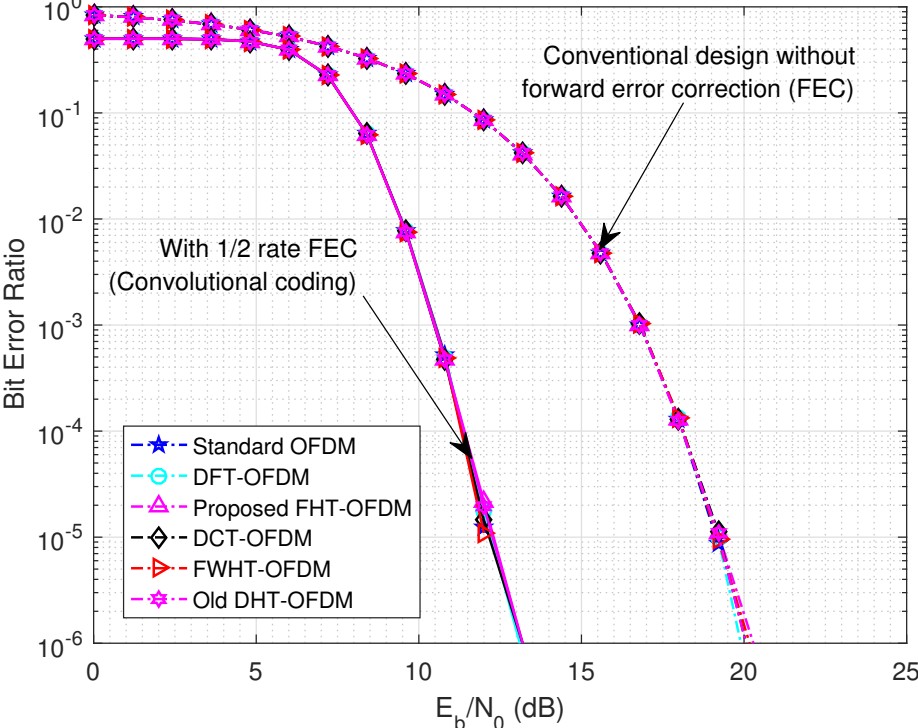

**Figure 8.** BER results of precoded OFDM (with convolutional coding) over flat fading channel corrupted by AWGN ($\ell_h = 1$ channel tap).

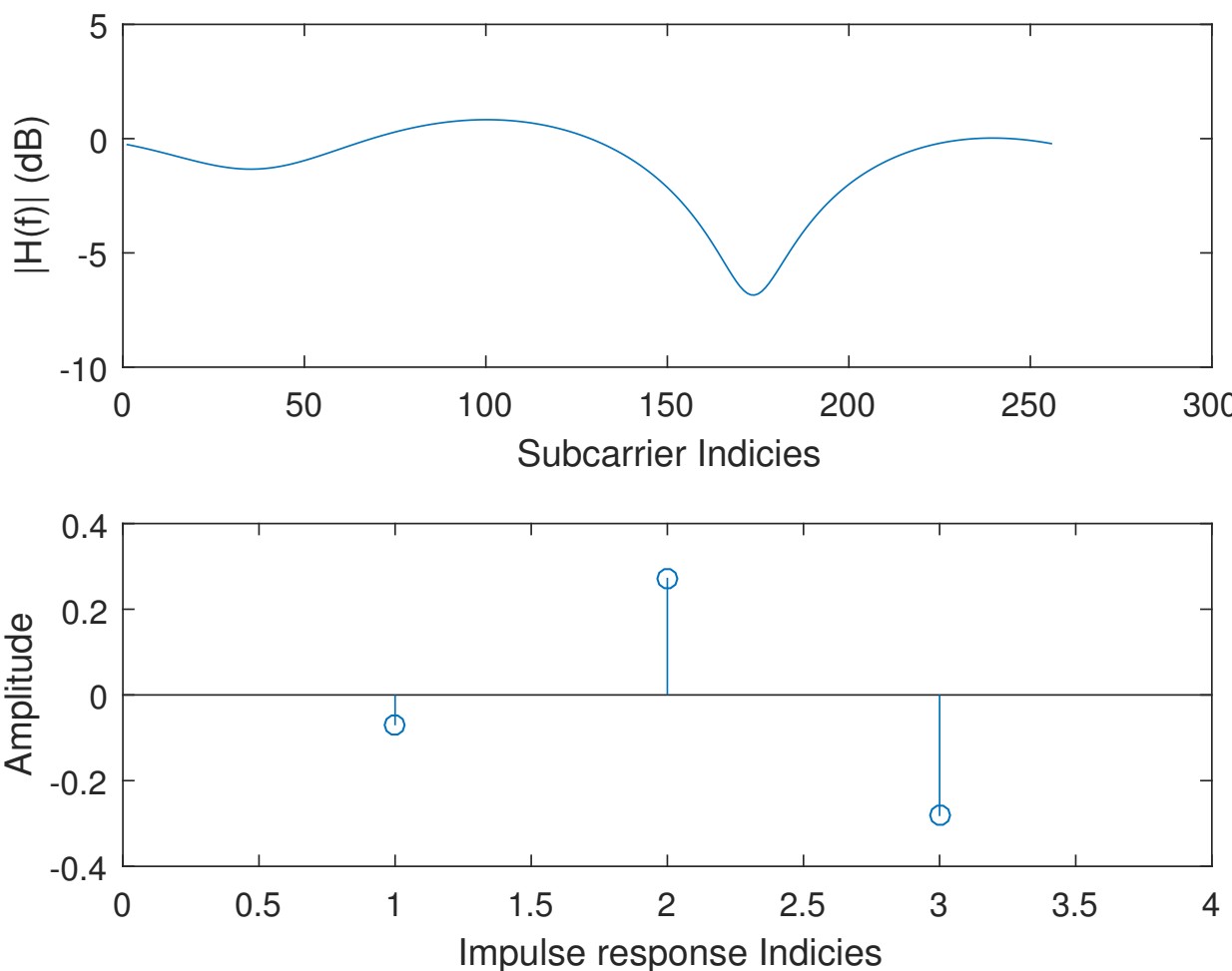

**Figure 9.** Channel and impulse responses ($\ell_h = 3$ channel taps).

*5.5. BER Performance with FEC with Interleaving*

In addition to the convolutional coding, we extend our results to include the performance of the proposed system with symbol interleaving. In this scheme, we apply the matrix interleaving earlier reported in [37,38]. In matrix interleaving, it involves an *n* input sequence that is transformed into $k \times m$ output sequence where $m = 16$ is the depth of the interleaver and $k = n/m$. Interleaving is required to remove burst errors due to impulse noise and fades and are usually used with convolutional or block error correction codes [37]. Over AWGN-only channel which induces a non-burst noise, the interleaving has no effect since the noise impact is only Gaussian distributed [37]. However, in this study which involves multipath fading channel having impulse responses, it can be seen that when interleaving is applied, the BER performance is improved as shown in Figure 11.

Clearly, all the precoded OFDM with FEC and interleaving outperform the non-FEC with no interleaving. However, comparing the results in Figures 10 and 11, for FEC with and without interleaving, respectively, it can be found that the precoded-OFDM with both FEC and interleaving outperform others having no interleaving by more than 1.2 dB at $10^{-5}$ BER.

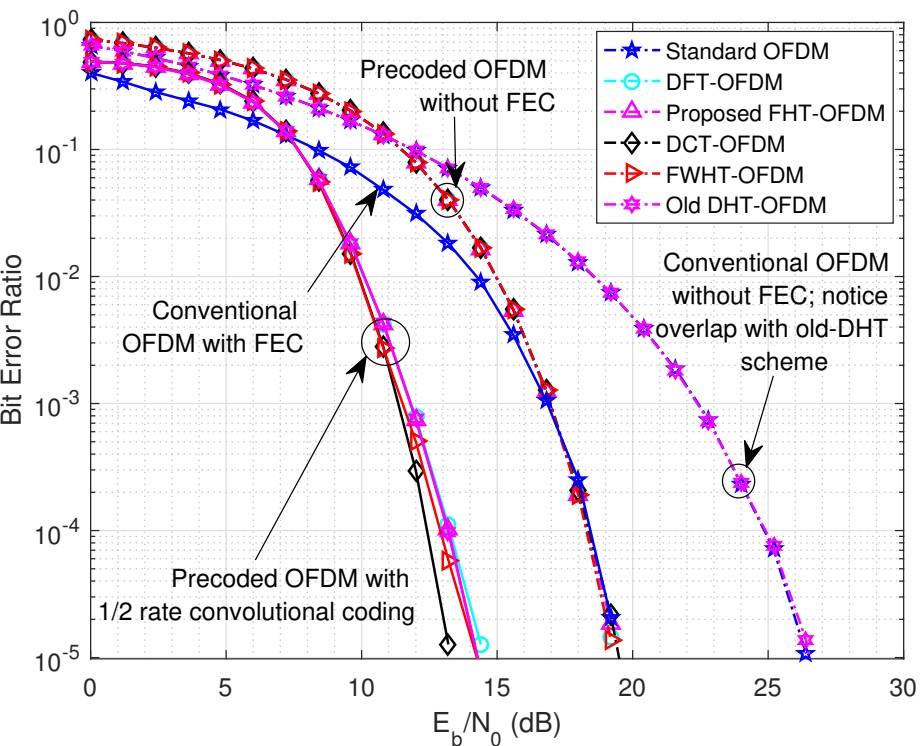

**Figure 10.** BER results of precoded OFDM (with convolutional coding) over frequency-selective fading channel corrupted by AWGN ($\ell_h = 3$ channel taps).

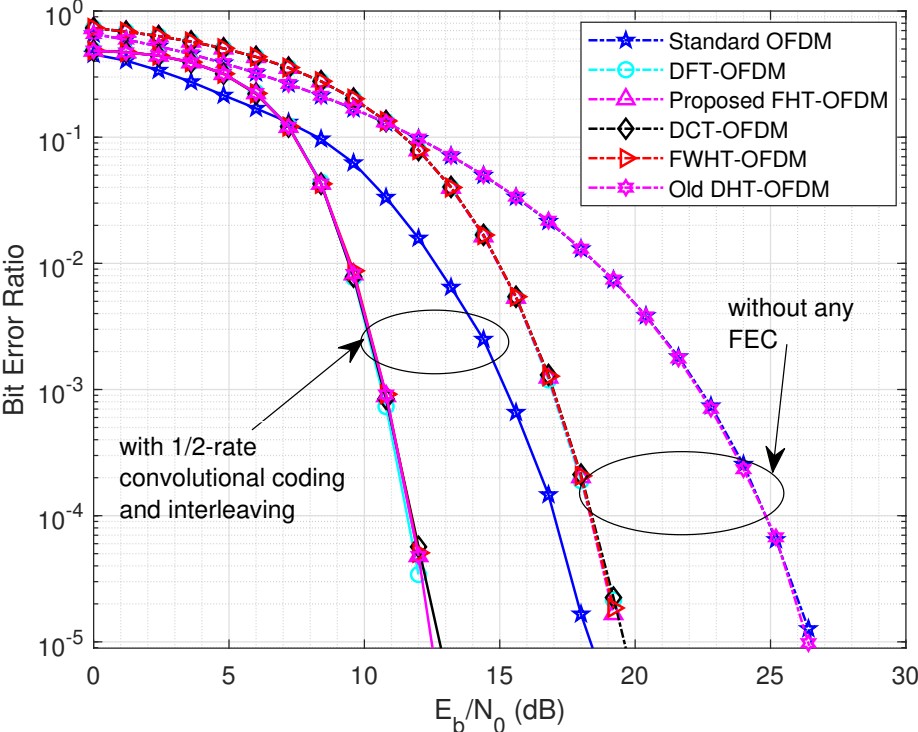

**Figure 11.** BER performance of DHT-precoded OFDM scheme (with convolutional coding and block interleaving) and other linear precoding schemes over frequency-selective fading channel ($\ell_h = 3$ channel taps).

## 6. Conclusions

In this study, we have presented an efficient algorithm for reducing the complexity of DHT precoding OFDM systems (to reduce the PAPR) for both transmitter and receiver sides. We showed that DHT can be expressed in terms of DFT, and by exploiting the unitary property of DFT matrices, that the DHT and IDFT can be simplified into as a single processing block reducing the system complexity. The proposed low-cost DHT precoding for OFDM schemes reduces the $\mathcal{O}(N^2)$ and $\mathcal{O}(N\log_2 N)$ complexities of DHT and DFT to a constant, respectively, in both transmitter and receiver sides. This could provide up to 74% power savings in the hardware systems. It was found that the proposal significantly reduces the PAPR of an OFDM scheme by 6.2 dB at $10^{-4}$ CCDF without compromising the BER; this increases power amplifier efficiency from 4.16% to 16.56%. The BER which was not reported earlier (in [19]) is also presented in this study which perfectly match that of the OFDM scheme. This makes the use of DHT-based precoding attractive for future mobile communication system standards such as 6G while minimizing the computational complexity, hardware size and communication latency. In addition to DHT and DFT performing better than other precodings in terms of PAPR, both of them are faster than all other schemes due to the reducedcomplexity from combining DHT-IFT and DFT-IDFT blocks, respectively. We also find that in harsh multipath fading environments, the FHT-precoded OFDM system outperforms the conventional OFDM system that is not precoded, whether the conventional OFDM and FHT-precoded OFDM systems are, individually, operated with error correction coding or not. With interleaving, the forward error-corrected precoded-OFDM system achieves additional 1.2 dB over FEC-only systems.

**Author Contributions:** Conceptualisation, K.A.; methodology, K.A. and M.V.R.; software, K.A.; validation, C.T., B.A. and C.H.S.; formal analysis, K.A. and M.V.R.; investigation, C.T.; resources, B.A.; data curation, K.A.; writing—original draft preparation, K.A.; writing—review and editing, C.T. and M.V.R.; visualisation, K.A. and C.H.S.; supervision, B.A.; project administration, K.A.; funding acquisition, B.A. and M.V.R. All authors have read and agreed to the published version of the manuscript.

**Funding:** This work was supported, in part, by the EPSRC " Community Peer-to-Peer Energy Trading and Sharing-3M (Multi-times, Multi-scales, Multi-qualities)" project Grant no. EP/N03466X/1, in part by the European Commission "Triangulum" (part of H2020 Smart Cities and Communities programme) project under Grant 646578-Triangulum-H2020-2014-2015/H2020-SCC-2014. In addition, this research was supported in part by Coordenação de Aperfeiçoamento de Pessoal de Nível Superior (CAPES) under Grant 001, Conselho Nacional de Desenvolvimento Científico e Tecnológico (CNPq) under grants 404068/2020-0 and 314741/2020-8, Fundação de Amparo à Pesquisa do Estado de Minas Gerais (FAPEMIG) under grants APQ-03609-17 and TEC-PPM 00787-18, and Instituto Nacional de Energia Elétrica (INERGE).

**Data Availability Statement:** My manuscript did not report any data.

**Conflicts of Interest:** The authors declare no conflicts of interest.

## Abbreviations

The following abbreviations are used in this manuscript:

| | |
|---|---|
| OFDM | orthogonal frequency-division multiplexing |
| PAPR | Peak-to-Average Power Ratio |
| IDFT | Inverse discrete Fourier transform |
| DFT | discrete Fourier transform |
| DHT | discrete Hartley transform |
| AWGN | Additive White Gaussian Noise |

| | |
|---|---|
| SC | Single Carrier |
| NR | New Radio |
| LTE | Long-term Evolution |
| CSI | Channel State Information |
| MMSE | Minimum Mean Square Error |
| ZFE | Zero-forcing Equalisation |
| STE | Single-tap Equalisation |

**Appendix A**

Similar to the conventional DFT (except by using FFT), DHT requires $\mathcal{O}(N^2)$ computations. Its major limitation is its computational complexity and, of course, hardware resource wastage. In [19], DHT was represented in terms of $\mathbf{F}$ as

$$\mathbf{W} = \frac{1+j}{2}\mathbf{F} + \frac{1-j}{2}\mathbf{F}^\dagger. \tag{A1}$$

Since $\mathbf{W}$ is unitary, $\mathbf{W}^\dagger\mathbf{W} = \mathbf{W}\mathbf{W}^\dagger = \mathbf{I}$ [24]. By letting $\mathbf{W}$ be the precoding matrix, we set $\mathbf{W} = \mathbf{P}$. In terms of the IDFT of the vector $\mathbf{C}$, from (2) the DHT precoded OFDM symbol before inserting cyclic prefix (CP) can be expressed as

$$\mathbf{c} = \mathbf{F}^\dagger\mathbf{W}\mathbf{X}, \tag{A2}$$

where $\mathbf{c}$ in (A2) represents the resulting vectorial DHT-precoded OFDM symbol in time-domain.

Based on involutory property, ref. [19] proposed a low-complexity DHT-precoding scheme that combines DHT and IDFT into a single-processing block in oder to reduce the computational complexity of the precoded OFDM scheme. By substituting (A2) into (4), the received signal over an AWGN-only channel becomes (while letting $\mathbf{P} = \mathbf{W}$)

$$\mathbf{y} = \mathbf{F}^\dagger\mathbf{W}\mathbf{X} + \mathbf{z}. \tag{A3}$$

At the receiver, the precoded signal that can be recovered at the output of the DFT is obtained as

$$\hat{\mathbf{X}} = \mathbf{P}^\dagger\mathbf{F}\mathbf{y} = \mathbf{P}^\dagger\mathbf{F}\mathbf{F}^\dagger\mathbf{P}\mathbf{X} + \mathbf{z}' \tag{A4}$$

where $\mathbf{z}' = \mathbf{P}^\dagger\mathbf{F}\mathbf{z}$ is the noise part after demodulation and removal of precoder. As $\mathbf{P}^\dagger$ and $\mathbf{F}$ are both unitary, their product is also unitary, thus $\mathbf{z}'$ remains Gaussian [24].

Since $\mathbf{W}$ can be expressed in terms of $\mathbf{F}$, let us observe closely the precoded signal part as follows:

$$\mathbf{c} = \mathbf{F}^\dagger\mathbf{W}\mathbf{X} = \left(\frac{1+j}{2}\mathbf{F}^\dagger\mathbf{F} + \frac{1-j}{2}\mathbf{F}^\dagger\mathbf{F}^\dagger\right)\mathbf{X} \tag{A5}$$

Based on the unitary property of matrix $\mathbf{F}$, we note that $\mathbf{W}^\dagger\mathbf{F} = \mathbf{I}$ and $\mathbf{W}^\dagger\mathbf{F}^\dagger$ is a flip matrix of the form

$$\mathbf{J}_{N \times N} = \begin{pmatrix} 1 & 0 & \cdots & 0 & 0 \\ 0 & 0 & \cdots & 0 & 1 \\ 0 & 0 & \cdots & 1 & 0 \\ \vdots & \vdots & \ddots & \vdots & \vdots \\ 0 & 1 & \cdots & 0 & 0 \end{pmatrix}. \tag{A6}$$

Although the approach reduces the complexity, due to (A6), the scheme is not energy efficient especially for hand-held battery-limited communication systems. Then, by substituting (A6) in (A5) we state that

$$\mathbf{X} = \left(\frac{1+j}{2}\mathbf{I} + \frac{1-j}{2}\mathbf{J}\right)\mathbf{X}, \tag{A7}$$

which is the result of applying DHT followed by IDFT on $\mathbf{X}$. Clearly, (A6) requires $\mathcal{O}(N)$ complexity. The signal from (A7) can now be processed as in the OFDM scheme for detection at the receiver. Although not treated in [19], let

$$\mathbf{G} = \mathbf{F}^{\dagger}\mathbf{W} = \left( \frac{1+j}{2}\mathbf{I} + \frac{1-j}{2}\mathbf{J} \right) \tag{A8}$$

in (A7), we show that the unitary property of $\mathbf{G}$ enables easy recovery of transmitted signal as

$$\hat{\mathbf{X}} = \mathbf{G}^{\dagger}\mathbf{y} = \mathbf{G}^{\dagger}(\mathbf{GX}) + \mathbf{Z}^{\dagger} \tag{A9}$$

where $\mathbf{G}^{\dagger}\mathbf{G} = \mathbf{I}$ and $\mathbf{Z}^{\dagger} = \mathbf{G}^{\dagger}\mathbf{z}$; since $\mathbf{G}$ is unitary, $\mathbf{Z}^{\dagger}$ is Gaussian. With matrix $\mathbf{J}$, the $\mathcal{O}(N^2)$ order matrix complexity is not well-reduced, making the design model power inefficient and therefore less desirable for battery operated and IoT-based modern devices.

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
