# Peer review of "On the Fast DHT Precoding of OFDM Signals over Frequency-Selective Fading Channels for Wireless Applications"

_electronics, doi:10.3390/electronics11193099_

Round 1

Reviewer 1 Report

This is an interesting research topic and the article is relevant and significant to the wireless communication community.

The authors are advised to make the introduction more robust by providing a more detailed technical narrative of what has informed the main contributions of the paper - consider using the current literature.

Also, the main contributions can be bullet-pointed for better presentation.

Reviewer 2 Report

This paper investigates the performance of combined OFDM and Discrete Hartley transform coding.

The paper is generally well written and presented, though there are some minor English issues as noted below.

The basic point of the paper is that the authors show that in some circumstances the inverse fast Fourier transform  (IFFT) and the discrete Hartley transform (DHT) can cancel out the effect of each other. This means that the transmitter is in-effect generating a time domain waveform similar to single carrier frequency division multiplexing (SC-FDMA) as used in the uplink of long term evolution (LTE) cellular systems. The contribution of this paper seems rather basic to me. However, it is not clear that this observation can easily be found in the existing literature. I think the material is well written and should be published after some revision.

I have the following comments and questions:

1. In the author's proposal, N multiplies are needed at the transmitter and receiver to multiply by (1+j)/sqrt(2) or (1-j)/sqrt(2). Since this is a constant phase shift of all data symbols, one could just neglect this phase rotation altogether? Alternatively, one can directly incorporate this phase shift in the modulation mapping to remove the N complex multiplies at the transmitter/receiver. 

2. As noted above, it seems that the authors have shown that effectively the combination of IFFT and DHT result effectively cancel each out out. This leads to a single carrier time domain waveform, e.g. as explained in ref [A] below. I think this point should be made much more clearly in the manuscript. This should be done in the abstract, introduction and conclusions at least. 

3. Fig 6 shows some results for processor energy. These comparisons are interesting, through realisticially at the transmit side, the power consumption of the power amplifier is typically much higher than these values.

3. I think the text of the paper could be reviewed to improve the presentation. Please address the following points:

(a) Page 1 - "using the involuntary property that is shared between DFT and IDFT" - I think "involuntary" property is the wrong term. I could not find this phrase in ref [9], which seems to be the author's source for this. Perhaps they can check again what is meant here.

(b) Page 2 - "same as the one reported in [9] although the BER performance of the received signal was not provided in that study" - The authors of [9] published a fuller version in ref [B] below which does include bit error rate results.  So please revise the text here.

(c) Page 8 - "In other words, the maximum achievable data rates is attained when the total transmission power is equally shared among the subcarriers, which is a very interesting result." - This result is also well known for the  waterfilling method when applied to multiple orthogonal channels that operate at high signal-to-noise ratio. So perhaps this is not totally unexpected.

(d) Page 15 - The authors argue that the J matrix in equation (A8) does not need to be implemented due to complexity issues. Similar to point 1 above, the permutation present in J could easily be applied in the transmitter and receiver without using order (N^2) multiplies as described in the text. Again though, this permutation is trivial in this reviewer's opinion and can easily be neglected as recommended by the authors. 

(e) Some minor English issues:

Page 2 - "spectral null problem and reduce PAPR" -> "spectral null problem and reduce the PAPR".

Page 2 - "to that of an SC" -> "to that of an SC waveform"

Page 2 - "transforms share similarly kernels" -> "transforms share similar kernels" 

Page 2 - "DHT kernel appeals to real-signals" -> "DHT kernel applies to real-signals"

Page 5 - "Since FFT implements" -> "Since the FFT implements"

Page 5 - the end of the first line of page 5 is badly formatted and goes past the end of the line. Please rephrase the text here to correct it.

Page 5 - "fizzles out" -> "reduces"

Page 11 - "These BER performances" -> "These BER performance results" (Note that "peformance" should always be singular in technical writing). 

Page 11 - "scheme are in two folds" -> "scheme are two fold"

Page 12 - "a fast-performing algorithm" -> "an efficient algorithm"

These are just examples of obvious errors and the paper should have a thorough proof-check to remove typos and awkward English phrasings.

References

[A] Grammenos, RC; Darwazeh, I, "SC-FDMA and OFDMA: The two competing technologies for LTE", In Proceedings of the Fourth International Symposium on Broadband Communication - ISBC'10. UTeM: Melaka, Malaysia

URL: https://discovery.ucl.ac.uk/id/eprint/1529403/

[B] Xing Ouyang,Jiyu Jin,Guiyue Jin,Peng Li, "Low Complexity Discrete Hartley Transform Precoded OFDM System over Frequency-Selective Fading Channel", ETRI journal 2015.

URL: https://onlinelibrary.wiley.com/doi/full/10.4218/etrij.15.0114.0513

Reviewer 3 Report

1.Improve the quality of the text presentation.

2. Add more recent year published papers as reference  and make compare the results with exited results.
